# Patterns of Tadpole β Diversity in Temperate Montane Streams

**DOI:** 10.3390/ani14081240

**Published:** 2024-04-20

**Authors:** Da Kang, Zijian Sun, Jiacheng Tao, Yan Huang, Tian Zhao

**Affiliations:** 1Key Laboratory of Southwest China Wildlife Resources Conservation (Ministry of Education), College of Life Science, China West Normal University, Nanchong 637009, China; kangda1013@126.com; 2College of Fisheries, Southwest University, Chongqing 400715, China; sunzj19@outlook.com (Z.S.); m18981366632@163.com (J.T.); 3CAS Key Laboratory of Mountain Ecological Restoration and Bioresource Utilization, Ecological Restoration Biodiversity Conservation Key Laboratory of Sichuan Province, Chengdu Institute of Biology, Chinese Academy of Sciences, Chengdu 610041, China

**Keywords:** amphibian, montane ecosystems, microhabitat variables, turnover, nestedness

## Abstract

**Simple Summary:**

Beta diversity is considered to be more accurate in reflecting the dynamics of community structure, as well as community assembly rules. However, many previous studies were only conducted in islands and lakes, while more attention is still needed for montane ecosystems. The main objective of the present study was to understand tadpole β diversity in temperate montane streams. The field work was conducted in 18 streams of Mount Emei, southwestern China, in 2018 and 2019. Our results indicated a high total tadpole β diversity, which was mainly contributed by a turnover process, and this pattern was shaped by both spatial and environmental factors.

**Abstract:**

Understanding the spatial variation and formation mechanism of biological diversity is a hot topic in ecological studies. Comparing with α diversity, β diversity is more accurate in reflecting community dynamics. During the past decades, β diversity studies usually focused on plants, mammals, and birds. Studies of amphibian β diversity in montane ecosystems, in particular, tadpoles, are still rare. In this study, Mount Emei, located in southwestern China, was selected as the study area. We explored the tadpole β diversity in 18 streams, based on a two-year survey (2018–2019). Our results indicated a high total β diversity in tadpole assemblages, which was determined by both turnover and nestedness processes, and the dominant component was turnover. Both the total β diversity and turnover component were significantly and positively correlated with geographical, elevational, and environmental distances, but no significant relationship was detected between these and the nestedness component. Moreover, the independent contributions of river width, current velocity, and chlorophyll *α* were larger than that of geographical and elevational distance. Overall, tadpole β diversity was determined by both spatial and environmental factors, while the contribution of environmental factors was larger. Future studies can focus on functional and phylogenetic structures, to better understand the tadpole assembly process.

## 1. Introduction

Understanding the spatial variation and the formation mechanism of biological diversity is a hot topic in recent ecological studies [1,2]. Biodiversity is typically classified into three components, including alpha (α), beta (β), and gamma (γ) diversity [3]. Specifically, α diversity refers to the species richness in a given area or community. γ diversity represents the sum of species richness in multiple communities, while β diversity represents the changes in species composition between different communities [4]. Therefore, β diversity can provide some compensative information to understand the dynamics of community structure, and thus has been widely used to examine the spatial and temporal patterns of diversity at both regional and global scales [5,6]. During the past decades, various approaches have been proposed to quantify β diversity, such as ploidy partitioning, additive partitioning, and similarity (or dissimilarity) indices [7]. These indices can be used to delineate biogeographic regions [8,9], as well as to conduct the siting of protected areas and the setting up of a network of protected areas [10]. However, these indices cannot explain the processes and mechanism of β diversity formation [11].

The existing studies suggest that the differences of species composition among communities can be generated by two main processes, which are species gain and loss and species replacement [12,13]. In order to quantify the contribution of these two processes to β diversity, Baselga decomposed the total β diversity (βsor) into nestedness (βsne) and turnover (βsim) components based on the Sørensen dissimilarity index [11]. Specifically, βsne reflects species gain and loss, and the underlying mechanism could be selective colonization, selective extinction, or dispersal limitation [14,15]. βsim indicates species replacement, which is mainly driven by environmental filtering, geographical barriers, or competition [16,17]. Since both of the two components play critical roles in determining the spatial distribution patterns of β diversity, they should not be split in studies [11,18,19,20]. For instance, Baselga found that the global latitudinal gradients in amphibian β diversity were determined by a combination of nestedness and turnover components [8]. Specifically, nestedness was the dominant component at high latitudes with abundant species richness, associated with the frequent occurrence of species additions and subtractions. In contrast, species substitution mainly occurred at low latitudes where species richness is relatively low, indicating the determination of a turnover process.

During the past decades, many other β diversity studies were conducted at the macro-ecological scale. Based on these results, ecologists tried to understand the general patterns of β diversity along latitudinal gradients [5,21]. These studies focused on various taxa, and were conducted in different areas across the world (e.g., bats and insects in the American continent [22,23], and mammals in North America [24]). These studies concluded that β diversity may exhibit a significant relationship with latitude. Based on these studies, it can be suggested that, similar to α diversity, the relationship between β diversity and latitude can also be summarized into three main patterns, including monotonically increasing, continuous decline, and non-significant relationships [21]. This is because β diversity patterns can be affected by many factors, such as the behaviors and life history traits of organisms per se, as well as external environmental conditions [21,25]. However, some other studies reported that the relationship between β diversity and latitude may be not significant, and thus they suggested the conducting of more research [26].

The spatial patterns of β diversity were also explored at the regional scale, and these studies typically focused on ecosystems such as islands and lakes. Despite ecologists focusing on different communities (e.g., bird, lizard, and invertebrate), they all found that spatial turnover contributed more to total β diversity than the nestedness component in island or lake ecosystems. Interestingly, this pattern can be determined by environmental variables, such as the size of the ecosystems, as well as the climate or microhabitat conditions [19]. However, studies related to communities’ β diversity patterns in montane ecosystems are still relatively rare. Indeed, mountains are important ecosystems maintaining more than 85% of the world’s species, and they are widely recognized as hotspots for the study of biodiversity and conservation [27,28]. More importantly, previous β diversity studies conducted in mountains mainly focused on plants [29], birds [19], and invertebrates [30]. We still need more studies that consider other taxa in mountains to better understand the general patterns of β diversity [20]. 

Among all the vertebrates, amphibians play critical functional roles in montane ecosystems. This is especially true for their larvae, tadpoles, which regulate the cycling of materials and energy flow in montane streams, and consequently strongly influence the ecosystem’s functioning [31,32]. Therefore, understanding the β diversity patterns of tadpole assemblages could provide important insights for maintaining the diversity and the stability of montane stream ecosystems. In the present study, we investigated the patterns of tadpole β diversity in temperate montane streams. Specifically, we (1) investigated the total dissimilarity, turnover, and nestedness components between tadpole assemblages; and (2) assessed the environmental determinants of tadpoles β diversity patterns. Since amphibian composition changes dramatically along an elevational gradient in montane ecosystems [33], we predicted that tadpole β diversity could be high. Based on tadpoles’ low mobility and migration capacity, we also predicted that turnover may be the main component. Moreover, based on a previous study showing that tadpole α diversity is determined by a combination of spatial and microhabitat variables [34], we predicted that tadpole β diversity patterns are also determined by both spatial and environmental factors. 

## 2. Materials and Methods

### 2.1. Study Area and Transects Selection

This study was conducted in the region of Mount Emei (29°16′–29°43′ N, 103°11′–103°37′ E), a transition zone between the Tibetan Plateau and the Sichuan Basin in southwestern China (Figure 1). The average annual temperature of this region is about 17.29 °C, and the average annual precipitation is about 1790 mm [35]. Vegetation cover can be classified into four types along an elevational gradient in this area. Those are, evergreen broadleaf forest (390–1500 m), mixed evergreen and deciduous broadleaf forest (1500–2100 m), mixed conifer–broadleaf forest (2100–2800 m), and cold–temperate coniferous forest (2800–3090 m) [36]. Based on our preliminary investigations, as well as the distribution and accessibility, we selected a total of 18 streams as the transects, covering an elevational gradient from 485 to 2865 m (Appendix A). These streams contained complex and diverse microhabitats, providing suitable habitats for different tadpoles. Moreover, all of the selected streams were permanent, to allow for continuous and long-term surveys. They were separated by deep mountain gorges and other prominent landmarks to reduce spatial autocorrelation.

### 2.2. Tadpole Sampling

Approximately 300 m length of each montane stream was selected and sampled for tadpoles. This is because these montane streams were long and topographically complex, with a distribution of countless tadpoles. In particular, some sections of these streams were blocked by waterfalls and cliffs, making them difficult to be fully surveyed [37]. Nonetheless, all the selected segments were suitable habitats for tadpoles, with a variety of different microhabitat conditions. Since tadpoles are nocturnal animals, sampling activities were conducted at night (between 20:30 and 23:00) using “D-frame” dipnets (opening diameter: 40 cm; depth: 35 cm; mesh size: 3 mm; RBF326, Renniaofei, Shijiazhuang, China) from May to June in 2018, with one stream being sampled per night. Specifically, two to three people walked at the edge of the streams from downstream to upstream at a pace of about 1 m/s, collecting individuals from all potential microhabitats for tadpoles, including rocks with or without vegetation and the water column from the surface to the bottom [38]. This approach can effectively prevent the escape of tadpoles to the downstream when they are facing a disturbance. We repeated the sampling processes between May and June in 2019 to obtain a stronger database, and the two years’ data were merged for further data analyses. It is worth mentioning that all collection activities were carried out in sunny weather conditions, as the rain can lead to poor water conditions in these montane streams, making it difficult for us to find tadpoles. All the tadpoles encountered were sampled and identified to species based on external morphology. For individuals that we could easily identified to species based on external morphology (e.g., *Quasipaa boulengeri* and *Leptobrachium boringii*), we recorded their numbers directly, and released them back to the original habitats. For those that could not be distinguished by morphological traits (e.g., *Megophrys shapingehsis* and *Megophrys omeomontis*), we took them back to the laboratory and used DNA barcoding analyses based on a fragment of the mitochondrial 16S rRNA gene for species identification. More details of the sampling protocols and the species identification processes can be found in Sun et al. [34].

### 2.3. Environment Variables

We first recorded the elevation, latitude, and longitude coordinates of each transect using the Ovi Map app (http://www.ovital.com, accessed on 1 May 2018). Then we selected 12 microhabitat variables that can potentially affect tadpole distribution and diversity based on previous studies [39,40], including water temperature (℃), water pH, water conductivity (μs/cm), dissolved oxygen (μmol/L), current velocity (m/s), substrate type, water depth (cm), river width (m), total phosphorus (mg/L), total nitrogen (mg/L), ammonium nitrogen (mg/L), and chlorophyll *α* (mg/L; Appendix A). Detailed measurements were as follows: we used portable instruments (Stra A, 520 M-01A, Thermi Fisher Scientific, Waltham, MA, USA) to measure the water temperature, water pH, water conductivity, and water dissolved oxygen. The current velocity was recorded at the upper of the water body by a portable current meter (LS1206B, Tongda, Huaian, China). Chlorophyll *α* was recorded by a water quality detector (Eureka Water Probes, Austin, TX, USA) directly. Substrate types were divided into four categories based on particle size and type (i.e., sand: <1 cm, gravels: 1–20 cm, rocks: >20 cm, and humus: mainly composed of leaf litter) [32]. River width and water depth were measured using a tape measure. Finally, water samples for each stream were collected in clean polyethylene bottles separately, preserved in a cool box in the field. They were brought back to the laboratory immediately and measured for total phosphorus, total nitrogen, and ammonium nitrogen in the Chengdu Institute of Biology, Chinese Academy of Sciences. These measurements and collections were repeated at 10 m intervals in each stream, and average values were used for further statistical analyses. All of these activities were carried out by the same person to ensure the accuracy of the data.

### 2.4. Statistical Analyses

Following Baselga [8,11], tadpole β diversity was calculated based on species presence and absence data in the transects. Specifically, total β diversity (i.e., βsor) was represented by the Sørensen dissimilarity index. The turnover component (i.e., βsim) was calculated as the Simpson dissimilarity index, while the nestedness (i.e., βsne) component was calculated by the difference between βsor and βsim. Because of the large number of transects (>15), average values of βsor, βsim, and βsne between pairwise transects were calculated to represent the total βsor, βsim, and βsne, separately. 

The β diversity indices between pairwise transects were calculated as follows:(1)$$\beta sor=\frac{b+c}{2a+b+c}$$
(2)$$\beta sim=\frac{min\;(b,c)}{a+min\;(b,c)}$$
(3)$$\beta sne=\beta sor-\beta sim=\frac{|b-c|}{2a+b+c}\times \frac{a}{a+min\;(b,c)}$$
where a is the number of species observed in both transects, b is the number of species that can be only found in one transect, and c is the number of species only distributed in the other transect. Finally, the ratio between βsne and βsor (i.e., βratio = βsne/βsor) was used to distinguish the main component that determined tadpole β diversity. Specifically, a βratio less than 0.5 indicated that total β diversity was mainly determined by the turnover component; otherwise, it meant that total β diversity was mainly contributed to by the nestedness component. βsor, βsim, and βsne were computed by the “beta.pair” function in the “betapart” package [41].

Before conducting the Mantel test analyses, we performed a Shapiro test to detect the normality of each environmental variable, and the correlations of pairwise environmental variables were tested using Spearman’s rank correlations. Based on our results, elevation and water temperature were significantly correlated with each other (*r* = 0.81, *p* < 0.01). We removed water temperature, as elevation was a more important filter affecting the distribution of amphibians [33,42]. This process was carried out via the “varclus” function in the “Hmisc” package [43]. After that, we used Mantel tests with 9999 permutations to examine the correlations between βsor, βsim, βsne and geographical, elevational, environmental distances, respectively. These were conducted by the “mantel” function in the “vegan” package [44]. Specifically, geographical distances were calculated based on the transects’ latitude and longitude data, using the “distm” function from the “geosphere” package [45]. Elevational distance was calculated by the elevational differences between pairwise transects. Environmental distances were measured as a Euclidean distance using standardized environmental variables (standard deviation = 1 and mean = 0) by the “vegdist” function in the “vegan” package [44]. After that, we used linear regressions to assess the relationships between multifaceted β diversity indices and geographical, elevational, and environmental distances, respectively. 

Moreover, we used multiple regression on the distance matrices (MRM) to explore the effects of geographical, elevational, and environmental distances on βsor, βsim, and βsne, respectively. Before conducting MRM analyses, we standardized each explanatory distance matrix using the “stdize” function of the “MuMIn” package [46]. We estimated the *p*-values of the MRM using 9999-times permutation tests [47]. Finally, we used hierarchical partitioning analyses to examine the relative contribution of each explanatory distance matrix to βsor, βsim, and βsne, respectively. These processes were conducted using the “hier.part” function of the “hier.part” package [48]. All calculations and analyses were performed in R 4.1.0 [49].

## 3. Results

During the two years of sampling, we collected a total of 1470 tadpoles, which belonged to 19 species from 6 families (Appendix A). Specifically, *L. boringii*, *Oreolalax omeimontis*, and *M. omeimontis* were the dominant species, accounting for 64.11% of the total number of individuals. Rare species were *M. shapingensis*, *O. major*, *O. schmidti*, *Microhyla fissipes*, *Amolops chunganensis*, *A. granulosus*, and *O. popei*, accounting for only 1.82% of the total number of individuals. Because no tadpoles were collected from three transects (i.e., Heishuicun, Shuangshuijing, and Jingding), they were excluded from further analyses.

### 3.1. β Diversity and Its Components

Tadpole total β diversity (βsor) was 0.78 ± 0.18 (mean ± SD). The turnover component (βsim) was 0.66 ± 0.28 (mean ± SD), and the nestedness component (βsne) was 0.12 ± 0.15 (mean ± SD; Figure 2). In addition, the βratio value was 0.16 ± 0.15 (mean ± SD).

### 3.2. The Influencing Factors of β Diversity and Its Components

Results of the Mantel test showed that both environmental and spatial variables were important factors in determining tadpole β diversity in the streams of Mount Emei (Table 1; Appendix A). Specifically, both βsor and βsim were significantly and positively correlated with geographical, elevational, and environmental distances (Figure 3). However, there were no significant relationships between the nestedness component and geographical, elevational, and environmental distances, respectively (Figure 3).

### 3.3. Relative Contribution of Independent Factors to Tadpole β Diversity

A large proportion of tadpole β diversity variability can be explained by the MRM models. For the total dissimilarity and turnover component, the MRM model explained more than 40% of the variability. In contrast, the MRM model only explained 22% of the variability of the nestedness component. Specifically, river width (Rw) had a positive effect on the total dissimilarity and turnover component, while water depth (Wd) had a positive effect on the nestedness component (Table 2).

Based on the hierarchical partitioning analyses (Figure 4; Appendix A), total dissimilarity was best explained by chlorophyll *α*, followed by river width and geographical distance. The turnover component was best explained by chlorophyll *α*, followed by river width and current velocity. In addition, the nestedness component was best explained by current velocity, followed by water depth and conductivity.

## 4. Discussion

The present study revealed a high level of total β diversity for tadpole assemblages in Mount Emei, which was mainly driven by the turnover component. This indicated a strong occurrence of tadpole replacement between montane streams. Our observations were consistent with previous β diversity studies for other taxa conducted in montane ecosystems showing the determination of the turnover component (e.g., amphibian adults [20]; ants [50]; and beetles [51]). Therefore, we argue that species turnover should be the main driver contributing to the β diversity spatial patterns in montane ecosystems. This is not surprising, as the climatic environment and habitat conditions change rapidly with the increasing of the elevation in these mountains. In the present study, *Fejervarya multistriata*, *M. fissipes*, and *Pelophylax nigromaculatus* were low-elevational specialists, while *M. shapingensis*, *O. major*, and *O. schmidti* were high-elevational specialists. This is because the distribution of these species was strongly associated with distinct external environments [34]. However, this is not the case for β diversity conducted in island ecosystems. For example, Zhou et al. found that the β diversity of ant assemblages in the Thousand Island Lake, China, was mainly contributed to by the nestedness component [52]. Similarly, nestedness also dominated the β diversity of butterfly assemblages in the Zhoushan Archipelago of China [53]. This is because these island ecosystems are close to each other with similar environmental conditions, which are created by the rise of water levels. Therefore, species specificity between communities was low, and the substitution of species occurred infrequently between these ecosystems. 

Geographical distance also had positive effects on tadpole β diversity in this study. Geographical distance has always been considered as one of the most important factors affecting the composition of biological communities [54]. Generally, communities showed a significant distance attenuation pattern with an increase in geographical distance (i.e., increasing geographical distance is accompanied by a decrease in the similarity of biological communities) [55]. This is because the variation in environmental conditions will increase with the increase in geographical distance [56]. Our results were consistent with a previous study showing that aquatic angiosperm β diversity was determined by geographical distance in China [57]. However, this pattern usually occurs in groups of organisms with a low mobility and migration capacity [58]. For those more-intense-dispersal-activity animals such as birds, geographical distance was not so important for determining community composition [59]. This may be because a large home range can compensate for the environmental differences between the habitats of different species [60].

Besides spatial factors, environmental distances were also crucial in generating the β diversity of tadpole assemblages. These results supported our initial prediction that tadpoles β patterns were determined by both spatial and environmental factors in montane streams. However, the relative importance of the two processes still requires further exploration. Specifically, the independent contributions of river width, current velocity, and chlorophyll *α* were larger than that of geographical distance and elevational distance. In our studies, some tadpoles were observed in streams with a wider river, low velocity, and high chlorophyll *α* (e.g., *P. nigromaculatus*, *M. fissipes*, and *F. multistriata*), while others may prefer to live in streams with a fast velocity, narrow river width, and low chlorophyll *α* (e.g., *L. boringii*, *O. omeimontis*, and *M. omeimontis*). Our previous study recognized that chlorophyll *α* was highly correlated with potential food resources for some specific tadpoles in montane streams (e.g., algae for *M. fissipes* and *F. multistriata* [34]), supporting their ability to breed and live in the ecosystems. Therefore, the two completely different types of microhabitat conditions supported totally different compositions of tadpole assemblages, and thus a high β diversity. This phenomenon can be observed in other animals distributed in mountains. For instance, the β diversity of aquatic insects in the Rocky Mountains of the southern United States was determined by the stream conditions, such as stream size and hydrologic connectivity [61]. The β diversity of tropical fish assemblages in the Bita River Basin in eastern Colombia was determined by environmental variables such as water parameters (e.g., conductivity and water temperature) and substrate type [62].

A large number of studies have shown that the formation of β diversity is the result of the integrated effects of two ecological processes, namely the niche process and neutral process [63,64]. The former holds that environmental differences (i.e., environmental filtering) are the main factors that lead to a change in community structure [64], while the latter emphasizes that dispersal limitations will lead to a change in species composition [65]. Our tadpole β diversity patterns were influenced by both spatial and microhabitat factors, supporting a previous study showing that both dispersal limitations and environmental filtering may play important roles in shaping tadpole assembly processes in montane streams [34,37]. However, functional traits and phylogenetical analyses are still needed to better understand the mechanism underlying the community assembly processes for tadpoles. Our results also supported a previous study suggesting that interspecific competition may play an important role in determining β diversity [66], as environmental filtering and dispersal limitations are associated with species interactions.

## 5. Conclusions

Overall, the present study investigated the patterns of tadpole β diversity in temperate montane streams. Our results indicated that tadpole total β diversity in the streams of Mount Emei was high, which was mainly contributed to by the turnover component. Interestingly, similar to tadpole α diversity, the β diversity patterns were shaped by both spatial and environmental factors. This supported previous claims that tadpole assembly processes were shaped by both dispersal limitations and environmental filtering. However, microhabitat features were more crucial in determining tadpole β diversity patterns. Our results also showed that a large proportion of β diversity indices (βsor, βsim, and βsne) was explained by neither environmental nor spatial distance, suggesting that other ecological processes, such as biotic interactions, may also affect tadpole β diversity. This can be verified in future studies. In addition, this study only considered taxonomic β diversity; functional and phylogenetic β diversity studies should be conducted to better understand the assembly process of tadpoles in montane streams. This study provides important information to better understand general β diversity patterns in mountains, as well as the conservation of amphibian diversity.

## Figures and Tables

**Figure 1 animals-14-01240-f001:**
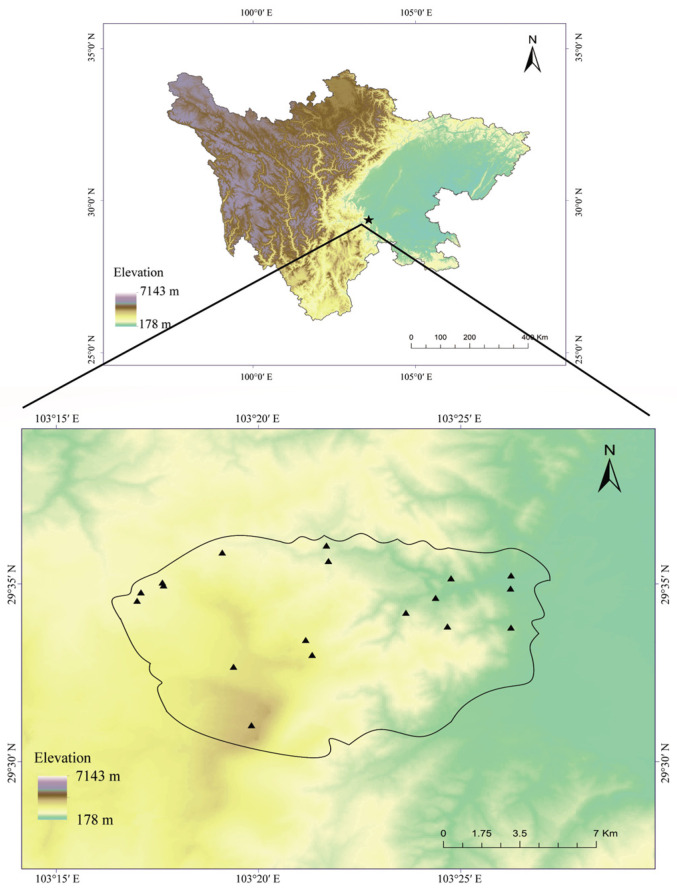
Map of the study area and the transects (The polygon represents the location of Mount Emei, and the triangles represent the positions of transects).

**Figure 2 animals-14-01240-f002:**
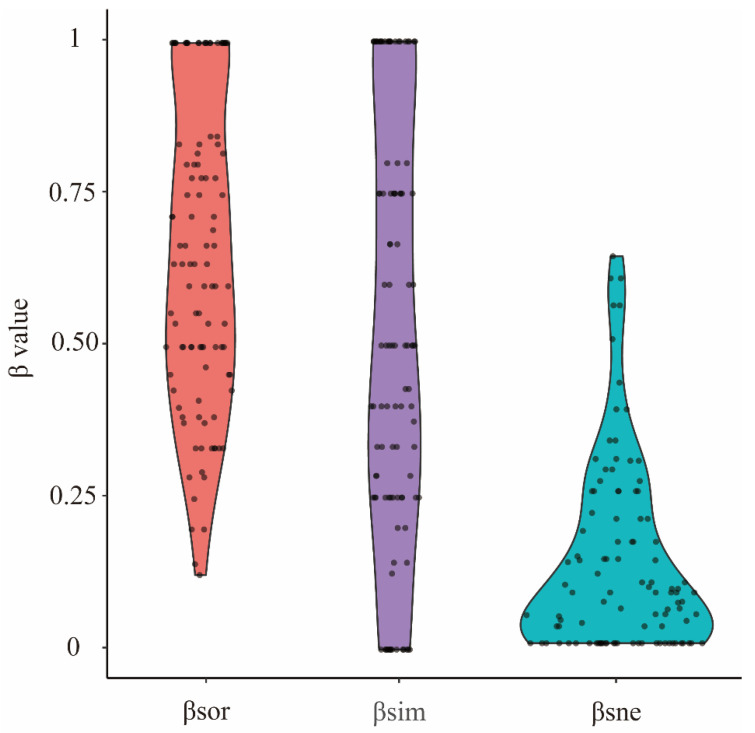
Tadpole β diversity indices in Mount Emei. βsor: total β diversity; βsim: turnover component; βsne: nestedness component.

**Figure 3 animals-14-01240-f003:**
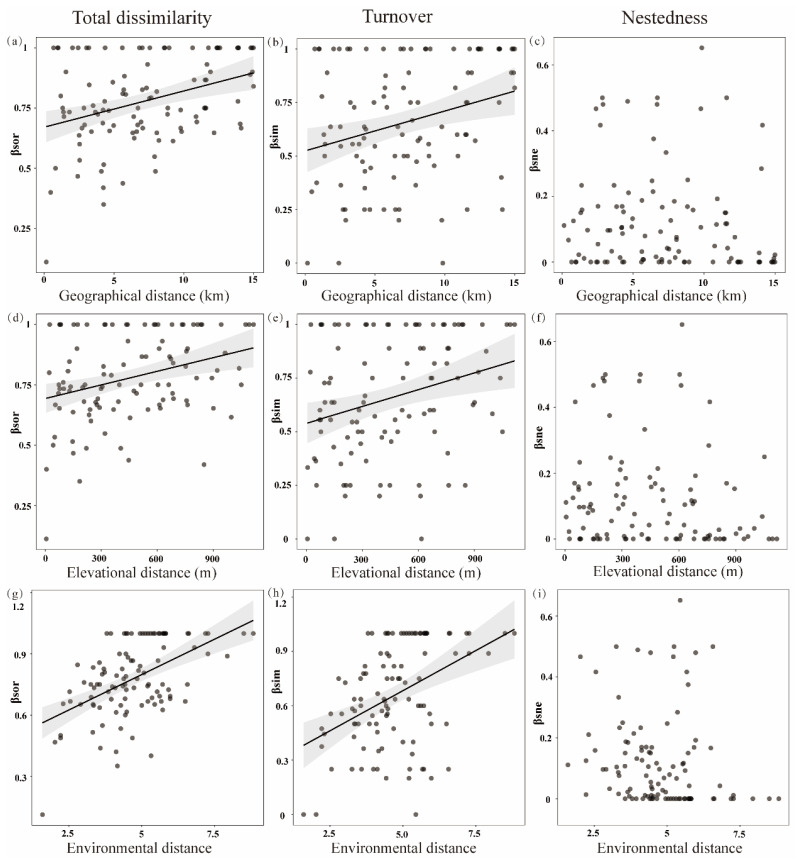
Relationships between multifaceted β diversity indices and geographical (**a**–**c**), elevational (**d**–**f**), and environmental distances (**g**–**i**). Solid lines indicate significant relationships (*p* < 0.05). Shallow area represents 95% confidence interval.

**Figure 4 animals-14-01240-f004:**
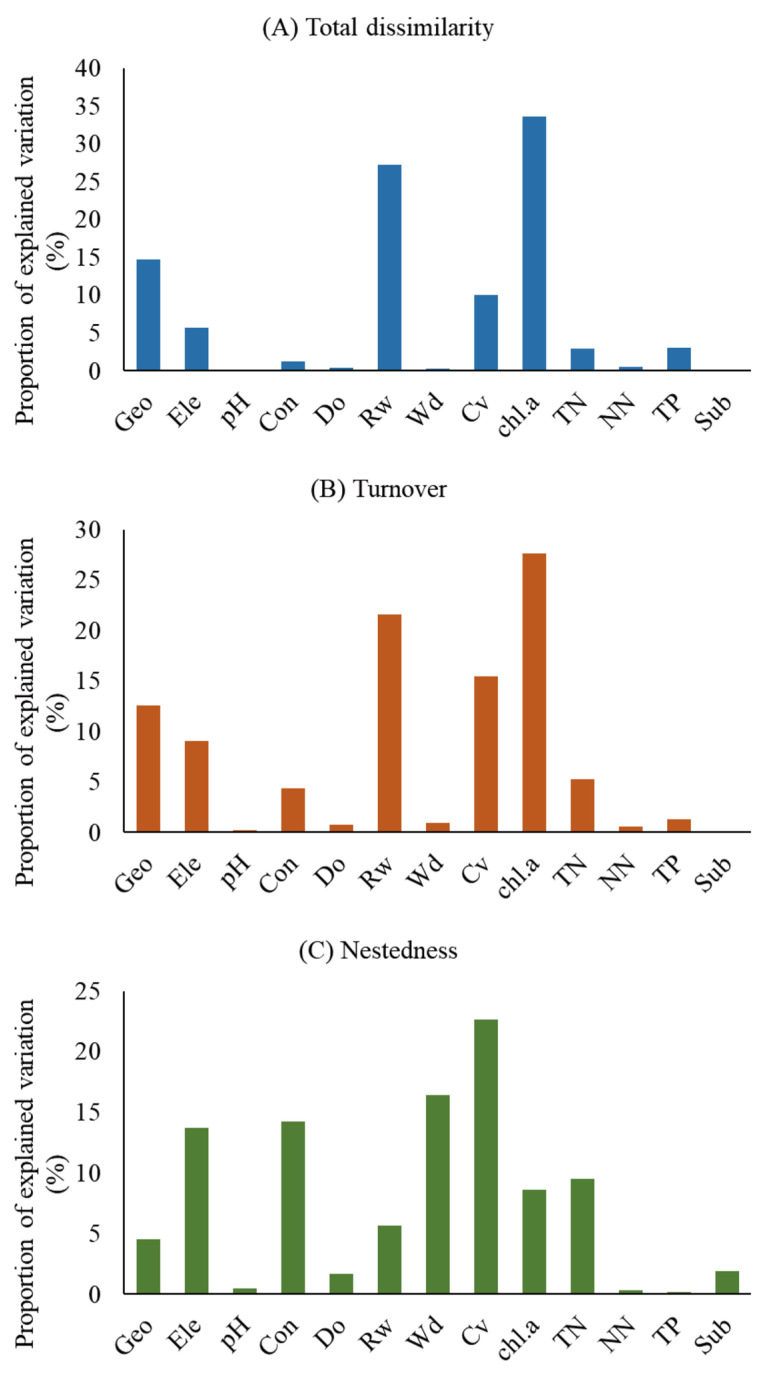
Independent contribution of each explanatory distance matrix in the variations of tadpole β diversity. The meaning of the abbreviations can be found in Table 2.

**Table 1 animals-14-01240-t001:** The correlation between multifaceted tadpole β diversity indices and geographical, elevational, and environmental distances. The bold values indicate *p* < 0.05. (Mantel tests were performed with 9999 permutations).

Distance Matrixes	Total Dissimilarity	Turnover	Nestedness
Geography	**0.007**	**0.008**	0.952
Elevation	**0.010**	**0.008**	0.962
Environment	**0.006**	**0.007**	0.959

**Table 2 animals-14-01240-t002:** The effects of environmental and spatial factors on tadpole β diversity.

Components	Total Dissimilarity	Turnover	Nestedness
R^2^	0.41	0.40	0.22
*p*	**0.05**	**0.04**	0.31
Geo	0.33	0.39	0.78
Ele	0.23	0.16	0.23
pH	0.96	0.86	0.61
Con	0.68	0.38	0.20
Do	0.58	0.62	0.83
Rw	**0.03**	**0.04**	0.29
Wd	0.82	0.39	**0.04**
Cv	0.41	0.13	0.09
chl.a	0.14	0.23	0.81
TN	0.39	0.25	0.25
NN	0.66	0.65	0.79
TP	0.29	0.45	0.96
Sub	0.88	0.78	0.43

Note: R^2^ represents the effect of the model that is explained by all variables. The partial regression coefficients (b) and associated *p* values of the model are obtained from the permutation test (9999 runs). The bold values indicate *p* < 0.05. (Abbreviations: Geo, geographical distance; Ele, elevational distance; pH, water pH; Con, conductivity; Do, dissolved oxygen; Rw, river width; Wd, water depth; Cv, current velocity; chl.a, chlorophyll *α*; TN, total nitrogen; NN, ammonium; TP, total phosphorus; Sub, substrate types).

## Data Availability

The datasets presented in this study are available from the corresponding authors on reasonable request. The data are not publicly available due to privacy restrictions.

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
