# Peer review of "Patterns of Tadpole β Diversity in Temperate Montane Streams"

_animals, 2024, doi:10.3390/ani14081240_

Round 1
Reviewer 1 Report
Comments and Suggestions for Authors
I read the manuscript with great interest. I only have a few comments below.
You found the tadpole beta diversity was strongly influenced by chlorophyll a. I wonder how that happened and it may be worth noting the underlying reason why in your Discussion section. Was this maybe because of potential food resources for certain species of tadpoles being high at locations of the chlorophyll (e.g., benthic invertebrates biomass being high)?
I am also curious how biotic interactions may have (or may not have) affected the beta diversity. I understand that this aspect was not in your present research scope, so it is okay that you did not cover it. But you can insert a few sentences describing the potential effects of competition and predation on the results you found. Was there any evidence of interspecific competition by closely related species not so important? Or any evidence of predation by animals such as birds or mammals being so insignificant? If beta diversity is not influenced by biotic interactions, then you can ignore this comment.
Thank you.
Author Response
Reviewer#1:
I read the manuscript with great interest. I only have a few comments below.
You found the tadpole beta diversity was strongly influenced by chlorophyll a. I wonder how that happened and it may be worth noting the underlying reason why in your Discussion section. Was this maybe because of potential food resources for certain species of tadpoles being high at locations of the chlorophyll (e.g., benthic invertebrates biomass being high)?
Reply: We totally agree that chlorophyll a is highly correlated with potential food resources for some specific tadpoles in montane streams. This has been provided in the Discussion section of the revised manuscript. (Line 318-321).
I am also curious how biotic interactions may have (or may not have) affected the beta diversity. I understand that this aspect was not in your present research scope, so it is okay that you did not cover it. But you can insert a few sentences describing the potential effects of competition and predation on the results you found. Was there any evidence of interspecific competition by closely related species not so important? Or any evidence of predation by animals such as birds or mammals being so insignificant? If beta diversity is not influenced by biotic interactions, then you can ignore this comment.
Reply: Interspecific competition could affect β diversity based on the limiting similarity hypothesis, as species with similar traits cannot coexist with each other in the same ecosystem. Therefore, and following Reviewer #1’s suggestions, this part has been provided in the Discussion section (Line 340-342).
Reviewer 2 Report
Comments and Suggestions for Authors
This paper investigates the patterns of tadpole beta-diversity in temperate streams of mountain regions. I think that the results and discussion of the paper are simple and clear, the data analyses are appropriate for scientific papers.
Here are some minor comments. I hope that the comments below will be helpful to the authors in their correction of the ms.
L102-105: The authors should somewhat mention that the prediction is based on low mobility and migration capacity. This corresponds with L304.
L160-163: I recommend that the authors provide averages and standard deviations of these microhabitat variables on supplementary. This will help the readers to understand the conditions of montane streams.
L161: The authors should show the layer (upper, middle or bottom) of the water body in which the current velocity was measured.
L230: Table S2 indicates that Odorrana margaretae was captured in Heishuicun. Is this right? If so, transects without tadpoles are not three but two. The authors should verify whether "further analyses" contained data of Heishicun.
L252-270: The numerical values presented in the text are almost identical to those shown in Table 2. The authors should avoid duplicating the same information in both the text and the tables as much as possible. It would be better to refine the representation in the text.
Comments on the Quality of English Language
L346: Is "not" necessary?
Author Response
Reviewer#2:
Comments and Suggestions for Authors
This paper investigates the patterns of tadpole beta-diversity in temperate streams of mountain regions. I think that the results and discussion of the paper are simple and clear, the data analyses are appropriate for scientific papers.
Here are some minor comments. I hope that the comments below will be helpful to the authors in their correction of the ms.
Reply: We appreciate this comment. We have carefully revised our manuscript following Reviewer#2’s suggestions. Details have been provided below:
L102-105: The authors should somewhat mention that the prediction is based on low mobility and migration capacity. This corresponds with L304.
Reply: Done (Line 104-105).
L160-163: I recommend that the authors provide averages and standard deviations of these microhabitat variables on supplementary. This will help the readers to understand the conditions of montane streams.
Reply: Following Reviewer #2’s suggestions, new table (Table S2) has been provided in the supplementary material.
L161: The authors should show the layer (upper, middle or bottom) of the water body in which the current velocity was measured.
Reply: Done (Line 166).
L230: Table S2 indicates that Odorrana margaretae was captured in Heishuicun. Is this right? If so, transects without tadpoles are not three but two. The authors should verify whether "further analyses" contained data of Heishicun.
Reply: We apologize for this mistake. We have carefully checked the results, and we are sure that no tadpole was captured in Heishuicun. The Table S3 has been updated in the revised manuscript.
L252-270: The numerical values presented in the text are almost identical to those shown in Table 2. The authors should avoid duplicating the same information in both the text and the tables as much as possible. It would be better to refine the representation in the text.
Reply: Done (Line 254-256, 267-271).
Comments on the Quality of English Language
L346: Is "not" necessary?
Reply: Not really. This word has been removed in the revised manuscript (Line 349).